# Underwater Pipeline Oil Spill Detection Based on Structure of Root and Branch Cells

**Huajun Song [1,2,*], Jie Song [1] and Peng Ren [1,2]**

[1]  College of Oceanography and Space Informatics, China University of Petroleum, Qingdao 266580, China; s17050647@s.upc.edu.cn (J.S.); pengren@upc.edu.cn (P.R.)

[2]  Key Laboratory of Marine Environmental Survey Technology and Application, Guangzhou 510000, China

*  Correspondence: huajun.song@upc.edu.cn

**Abstract:**  The existing oil spill detection methods mainly rely on physical sensors or numerical models cannot locate the spill position accurately and in time. To solve this problem, combining with underwater image processing technology, an unsupervised detection algorithm for oil spill in underwater pipelines is proposed for the first time. First, the oil spill region to be detected is regarded as the moving target, and the foreground detection algorithm is applied to the processed images. Then, the HSV (Hue, Saturation, Value) color space of the image is used to screen the oil spill region meeting the threshold requirements. Next, the bitwise of foreground mask and HSV mask into cells are divided. Finally, according to the characteristics of the oil spill image, false detection  is eliminated by classifying cells three times. After qualitative and quantitative analysis, it is proved that the proposed algorithm can detect oil spill region accurately.

**Keywords:** object detection; underwater image; computer vision; oil spill

## 1. Introduction

There have been several oil spills at sea over the decades, e.g., the Gulf of Mexico (2010), the Timor Sea (2009), and the North Sea (1988). Although oil spills occur infrequently, they cause severe economic losses and a substantial adverse impact on the global environment [1]. Take the oil spill in the Gulf of Mexico in 2010 as an example: the accident caused eleven deaths and economic losses amounted to tens of billions of dollars. About 757,000 m$^3$ of oil spilled, which seriously threatened the marine ecological environment.

To provide technical support for environmental risk assessment, emergency plan formulation and emergency response, researchers have carried out numerous oil spill detection studies. The existing oil spill detection methods can be roughly divided into two directions according to detection location.

Above the sea surface, most of them are detected by synthetic aperture radar (SAR) satellite images or physical sensors. Dhont proposed the combined use of SAR and underwater gliders for oil seeps detection [2]. However, due to poor timeliness, leakage points cannot be found in time and the cost is high.

The other direction is underwater which mainly focuses on the numerical model research. Øistein regarded the discharges as terms of multiphase plumes and proposed a plume model based on Lagrange [3–5]. With image processing technology development, many target detection and tracking algorithms based on image features have been proposed. Compared with numerical models, vision-based target detection algorithms are more versatile. Some scholars have carried out image-based oil spill research. Bescond proposed a photoacoustic detection and monitoring method of oil spill [6]. Osman estimated the oil spill flow rate base on Wavelet in optical images [7]. Qasem implemented an underwater remotely operated vehicle (ROV) and identified oil spills by

binarized difference images [8]. Oh proposed an underwater multispectral imaging system for environmental monitoring to identify different oil samples [9]. However, the current use of image processing technology is based on the image of the simulation experiment, with a simple image background, which is different from the actual situation, and the primary purpose is to use the oil spill image which spill region is known to calculate the relevant values, such as oil spill rate, oil spill amount and oil spill area.

With the development of machine learning, it has been widely used in image processing. Using convolutional neural networks (CNN), general target detection algorithms have achieved good results on object recognition datasets such as Pascal VOC (Visual Object Classes) [10], COCO (Common objects in context) [11], and ImageNet [12], which consist of a large number of pictures and their annotated documents. However, target detection algorithms based on machine learning often require massive amounts of datasets to extract features of the target. Commonly used training sets often reach the level of millions or even hundreds of millions of samples. However, underwater oil spill images are challenging to obtain and scarce in practice, which cannot meet the requirements of statistical-based detection model training. Therefore, the target detection methods based on machine learning are not suitable for underwater oil spill detection task. Feature extraction has to adopt traditional methods.

Considering the oil spill movement characteristics are significant, motion detection can be applied in detecting the oil spill. Although there are various motion detection methods, none achieve good results in dynamic monitoring scenarios. Especially due to a lot of interference in the underwater environment, such as seawater and marine life movement, underwater image motion detection's difficulty is further increased. Thus, false detection is easy to occur, and the detection results need to be processed to reject false detection.

Considering the maturity of the existing algorithms and the limitation of their application on underwater oil spillage detection, an unsupervised detection algorithm based on underwater oil spill motion and color characteristics is proposed. It integrates the characteristics of the oil spill image to eliminate false detection by using the proposed cell structure. It achieved high accuracy in experiments. The rest of this paper is as follows. Section 2 introduces the proposed algorithm and its specific implementation. Section 3 presents the detection results and evaluation of the proposed algorithm. Section 4 gives the conclusion of the paper.

## 2. Method

The proposed algorithm processes the video images collected by underwater cameras. Firstly, theunderwater oil spill is regarded as the motion foreground, and the motion detection algorithm is used to detect it. The suspicious oil spill region is screened out by threshold in HSV color space too. Next, the two suspicious oil spill regions are united into the pre-detecting mask and the mask image is separated into cells, which can be classified as oil cells or others. The spatiotemporal context information is used in cells to eliminate missed targets and lock the oil leakage region. Finally, combining the plume characteristic of spilled oil, a structure of root and branch cell is proposed, and a cluster of cells belonging to the same root cell is regarded as the detected oil spill region.

Figure 1 is the flowchart of the proposed algorithm. Figure 2a shows the implement of the algorithm. Figure 2b shows the process of pre-detection. It unites motion detection and color screening in HSV space to generate a suspected oil spill region mask. Figure 2b–d are the single image feature detection based on cell structures. From left to right are cell divisions, root-branch classification, and root-branch stacks. Figure 2d is the process of eliminating false detection by introducing STC information into cells and the final detection result.

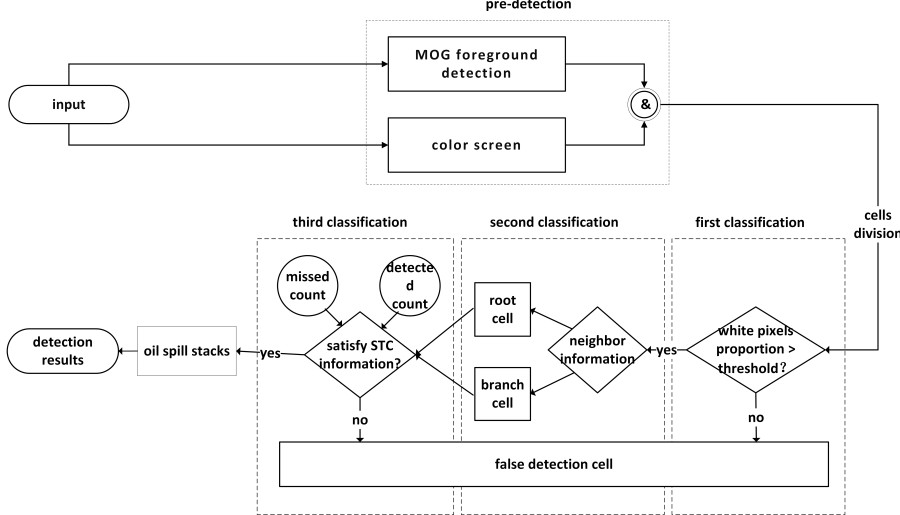

**Figure 1.** Algorithm flowchart.

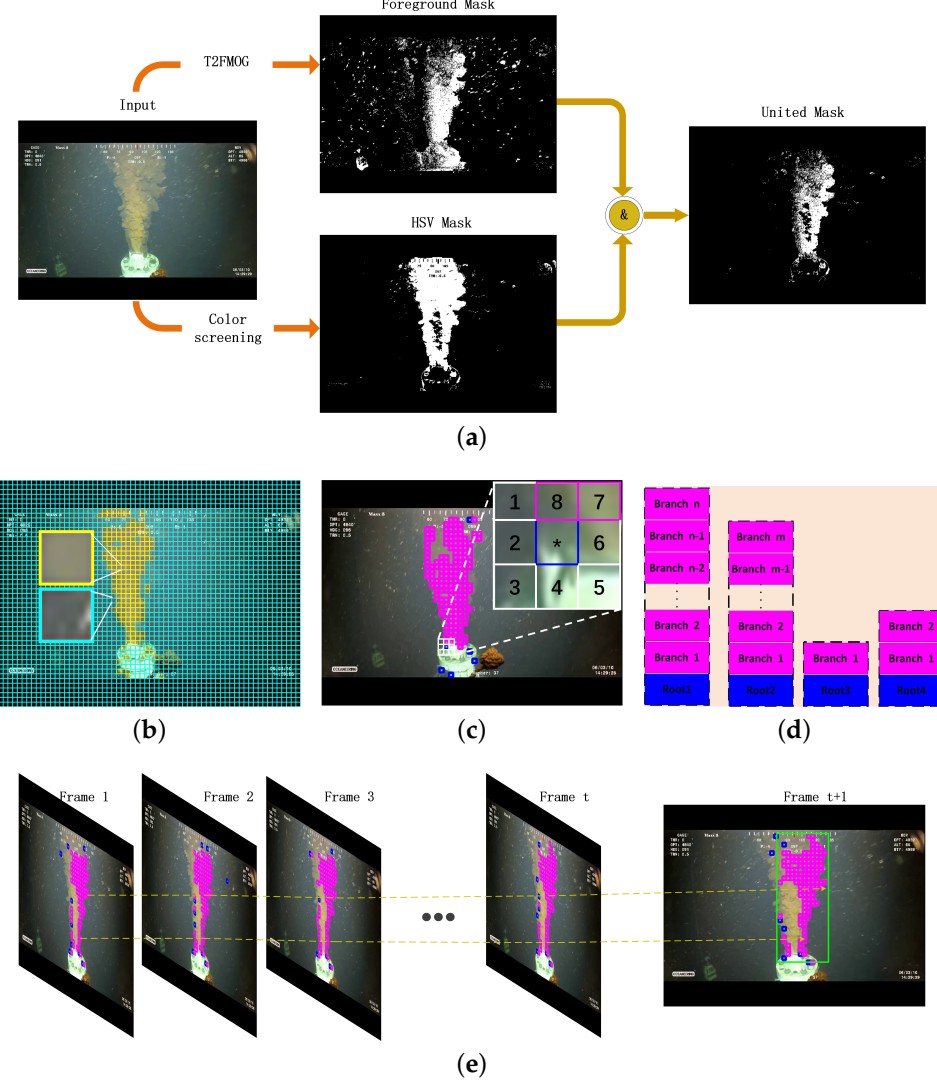

**Figure 2.** Algorithm implementation: (**a**) pre-detection; (**b**) cells division; (**c**) first classification; (**d**) second classification and (**e**) third classification.

In the past two decades, it is widely accepted that the progress of object detection has generally gone through two historical periods: "traditional object detection period (before 2014)" and "deep-learning-based detection period (after 2014)" [13]. Limited by the computing power and the lack of datasets, traditional target detection mostly uses handcrafted features, and then uses the classifier to classify the features. However, training a deep-learning-based model, such as SSD [14], Faster R-CNN [15], and YOLO [16], requires many data, thus oil spill detection tasks lacking underwater image information are not suitable for deep-learning-based detection models. To detect oil spill in underwater images, we use an unsupervised algorithm system combining multiple image features to achieve underwater oil spill detection.

*2.1. Foreground Detection*

Since the oil spill is continuously moving and changing, the underwater oil spill can be regarded as the moving target. The underwater oil spill detection can be regarded as the motion foreground detection task. Traditional motion detection algorithms include background modeling [17], clustering [18], image segmentation algorithm. However, it is difficult to detect an object in the dynamic background due to factors such as varying illumination, a sudden change in a scene, occlusion, and shadow [19]. Some researchers have applied machine learning methods to motion detection in recent years, such as FgSegNet series algorithms [20,21], which won first place in the CDNet2014 challenge [22]. However, these supervised algorithms are based on dataset training and have no universality, so they cannot apply to underwater oil spill detection.

However, looking at the literature, we can remark that there is often a gap between the current methods used in real applications and fundamental research [23]. After trials and contrasts, T2FMOG [24] based on a fuzzy mixture of Gaussians model is used in this paper, which has a higher tolerance to the background of slight motion and can produce ideal foreground detection effect in more complex underwater environment.

According to the uncertain mean vector or error, T2FMOG is divided into T2FMOG-UM and T2FMOG-UV. Due to the water flow in an underwater environment, the uncertainty variance is more likely to occur. Therefore, T2FMOG-UV can achieve better results. For RGB images, the multivariate Gaussian distribution with uncertain variance is as follows:

$$\eta(o, \mu, \tilde{\sum}) = \frac{1}{\sqrt{(2\pi)^3 |\Sigma|}} \prod exp(-\frac{1}{2} \frac{(X_{t,c} - \mu_c)^2}{\sigma_c}) \tag{1}$$

in which $\sigma_c \in [\underline{\sigma}_c, \overline{\sigma}_c]$, $c \in \{R, G, B\}$, each exponential component in Equation (1) is the Gaussian primary membership function (MF) with standard deviation. The upper MF is $\underline{h}(o) = f(o; \mu, \overline{\sigma})$ and the lower MF is $\overline{h}(o) = f(o; \mu, \underline{\sigma})$.

Between $\underline{h}$ and $\overline{h}$ is the footprint of uncertainty (FOU) as Figure 3. To control the intervals, a factor $k_v$ introduced:

$$\underline{\sigma} = k_v \sigma, \overline{\sigma} = \frac{1}{k_v}\sigma, k_v \in [0.3, 1] \tag{2}$$

Our experiments verify that the result is more accurate when $k_v = 0.9$. After getting the T2FMOG-UV model of the image, we can detect the image foreground. Since the background is more present than moving objects and its value is practically constant, a background pixel corresponds to a high weight with a weak variance. Based on this prior knowledge, K Gaussian models are sorted by $w_j/\sigma_j$ as a ratio. The first Gaussian distribution whose threshold exceeds T is regarded as the background distribution in Equation (3); other distributions are considered as foreground.

$$B = argmin_b(\sum_{i=1}^{b} w_{i,t} > T) \tag{3}$$

when a frame of the video comes in, use the log-likelihood and only consider the bilateral likelihood interval to test and classify each pixel:

$$H(X_t) = \left( \frac{1}{1/k_v{}^2 - k_v{}^2} \right) \frac{|X_t - \mu|^2}{2\sigma^2} \tag{4}$$

where $\mu$ and $\sigma$ are the mean and the std of the original T1 MF without uncertainty. If $H(x_t) < k\sigma$, the pixel is attributed to Gaussian distribution. Therefore, the pixel test results can be divided into two cases:

1.  If a pixel belongs to a certain Gaussian distribution, then the pixel belongs to the category represented by the Gaussian distribution.
2.  If a pixel is not attributed to a Gaussian distribution, then the pixel is attributed to the motion foreground.

Furthermore, to ensure the accuracy of the background model, the parameters update according to Equations (5) and (6).

Case 1: Matched Gaussian distribution

$$
\begin{aligned}
w_{i,t+1} &= (1 - \alpha)w_{i,t} + \alpha \\
\mu_{i,t+1} &= (1 - \rho)\mu_{i,t} + \rho X_{t+1} \\
\sigma_{i,t+1}^2 &= (1 - \rho)\sigma_{i,t}^2 + \rho(X_{t+1} - \mu_{i,t+1})(X_{t+1} - \mu_{i,t+1})^T
\end{aligned}
\tag{5}
$$

$\alpha$ is a fixed learning rate, $\rho = \alpha\eta(X_{t+1}, \mu_i, \Sigma_i)$. For unmatched distribution, $\mu$ and $\sigma$ are invariant, $w_{j,t+1} = (1 - \alpha)w_{j,t}$.

Case 2: The final probability distribution K

$$
\begin{aligned}
w_{k,t+1} &= Low\ Prior\ Weight \\
\mu_{k,t+1} &= X_{t+1} \\
\sigma_{k,t+1}^2 &= Large\ Initial\ Variance
\end{aligned}
\tag{6}
$$

Thus far, the motion foreground and the background model, which can be updated, are obtained by using T2FMOG. However, when considering factors such as water currents, marine life movement, and lens motion, there may be non-oil spilled parts in the foreground mask, so it is necessary to eliminate false detection by other methods.

*2.2. Color Screening*

In addition to the motion features, the oil spill region's color feature in an underwater environment is the most obvious. Therefore, the color threshold screening method is considered for screening the detected foreground further. However, in most videos, to display conveniently, the video pixel value is often in RGB format, which is designed according to the principle of color luminescence. The color is displayed by the excitation of red, green, and blue. Thus, it is not accurate enough for oil spill segmentation by using RGB color space. HSV is an intuitive color model, and its color parameters are hue, saturation, and lightness. Therefore, the videos can be converted to HSV space to screen the oil spill region under the foreground mask. The S and V channel calculation methods are: $S = V - min\{R, G, B\}$ amd $V = max\{R, G, B\}$, respectively. The channel calculation method of H is shown in Equation (7). When $H < 0$, $H = H + 360$.

$$
H = \begin{cases}
60(G - B)/(V - min(R, G, B)) & if\ V = R \\
60(B - R)/(V - min(R, G, B)) + 120 & if\ V = G \\
60(R - G)/(V - min(R, G, B)) + 240 & if\ V = B
\end{cases}
\tag{7}
$$

After obtaining the HSV model, the mask of the suspected oil spill region is obtained by threshold screening, and the screening method is as follows:

$$Mask_i = \begin{cases} 1, & \underline{T}_i \leq I_i \leq \overline{T}_i \\ 0, & others \end{cases} \tag{8}$$

where $\underline{T}_i$ and $\overline{T}_i$ are the lower and upper thresholds of each color channel, respectively. $I_i$ is the pixel value on a single color channel in HSV space. The pixel value on mask is 1 only when the pixel values on three channels meet the threshold requirements; otherwise, it is 0.

A single frame mask of the suspected oil spill region can be obtained by performing a bitwise operation between the foreground detection mask and color screened mask. However, there is still false detection in practical applications caused by those moving objects with the same color as spilled oil. Further operations are needed to eliminate false detection after obtaining the detection mask of a single image.

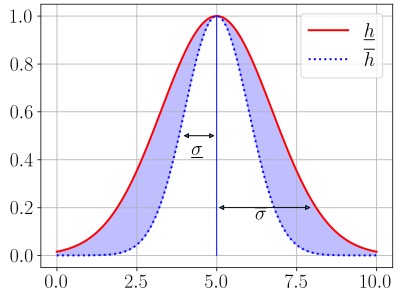

**Figure 3.** Gaussian membership function with uncertain variance.

## 2.3. False Detection Elimination

After watching more than 60 min of video containing oil spills, we found that the underwater oil spill image has two characteristics:

- Oil spillage is always produced from bottom to top.
- The movement of oil spillage occurs locally.

Based on these two characteristics, the root and branch cell structure and the STC method are proposed to eliminate false detection. The oil spill region is finally accurately screened out by classifying the cells three times.

Aiming at the first feature of underwater oil spill images, an algorithm of eliminating false detection based on cells with root and branch structures is proposed. First, the masked image is divided into S*S cells; then, the cells containing suspected oil spills are pre-classified and screened. The screening rule follows Equation (9), in which $O_i$ is the category of the ith cell, $W_i$ is the number of white pixels in the masked cell, and T is the set threshold, $0 < T < 1$. Figure 2b shows the screening diagram

$$Cell_i = \begin{cases} 1, W_i/S^2 \geq T \\ 0, W_i/S^2 < T \end{cases} \tag{9}$$

Then, the algorithm traverses the cells that have been classified as an oil spill from bottom to top and from left to right and secondarily classifies the pre-classified cells into root cells or branch cells. The classification method is shown in Figure 2c. The category of the cell is determined by the eight cells in the neighborhood. If there is no root cell or branch cell in the center cell neighbor positions 2–5, the center cell is classified as a root cell, and a new cell stack is established. If there is a root cell or branch cell in the center cell neighborhood, the center cell is pushed into the neighboring cell's

stack. Figure 2d shows the second classification method.The second classification method is shown in Equation (10), in which $neighbor_i$ is the ith neighbor cell category.

$$Cell_i = \begin{cases} root, \sum_{i=2}^{5} neighbor_i > 0 \\ branch, \sum_{i=2}^{5} neighbor_i = 0 \end{cases} \tag{10}$$

The movement of oil spillage occurs locally; that is, the movement amplitude of the oil spill in a short period is tiny, especially when the seawater movement is relatively gentle. The oil spill at the leakage point generally does not do translation movement. However, there is a diffusion movement at the edge part, and the internal motion amplitude is low or even relatively static. It is familiar to STC information, which models the statistical correlation between the low-level features (i.e., image intensity and position) from the target and its surrounding regions. Taking advantage of the oil spill movement characteristics, we propose an oil spill detection method that adds temporal and spatial context information to the cells. Due to the low amplitude of oil spill translation, the oil spill cell can always be detected within a certain period, and it cannot be detected for a long time when the target cell is an oil spill. The number of times each cell is detected is counted, and the cell is classified for the third time.

On the one hand, when the detection count is greater than the set value, the cell is confirmed to be an oil spill. On the other hand, when the continuous missed detection count is greater than a certain value, the detection count is set to zero, and the cell is regarded as a false detection for exclusion. The cell classification method of the tth frame is as follows:

$$Cell_t = \begin{cases} oil, \sum_{f=t-w}^{t} h_f > T_d \quad \& \quad \sum_{f=t-w}^{t} m_f < T_m \\ others, \sum_{f=t-w}^{t} m_f \geq T_m \end{cases} \tag{11}$$

where $w$ is the width of the counting window, $h_f$ is the category of the fth frame determined by Equation (9), and the judgment rule of $m_f$ is the opposite. $T_d$ and $T_m$ are detected and missed thresholds, respectively. The third classification process is shown in Figure 2c.

Finally, all cells are traversed again and popped up, taking the circumscribed rectangle of the graph formed by the cells in the same stack to form the detection boxes. The areas of the detection boxes are calculated, and the detection box with an area smaller than a certain threshold is regarded as invalid detection and excluded. The remaining detection boxes are used as the final detection result.

## 3. Results and Discussion

The dataset used in the experiment comes from underwater oil spill videos collected on the Internet, including some videos recorded by remote operated vehicle (ROV) during the Deep Horizon accidents. The videos were resized to a resolution of 640 × 480, and then cut into 1278 video frames.

Figure 4 shows the results of underwater motion detection [25]. The experimental results take the 60th and 215th frames of a video for comparison. Figure 4a shows the original frames. Figure 4b–d provided the detection results of several algorithms based on background modeling. It shows that the general background modeling algorithm cannot effectively distinguish the oil spill region and the seawater background, resulting in many missed detection. Figure 4e is the result of VIBE algorithm foreground detection, which can accurately detect part of the oil spill region, but there is still some missed detection. Figure 4f gives the detection results of the foreground detection algorithm used in this paper, which accurately detects oil spill movement in videos.

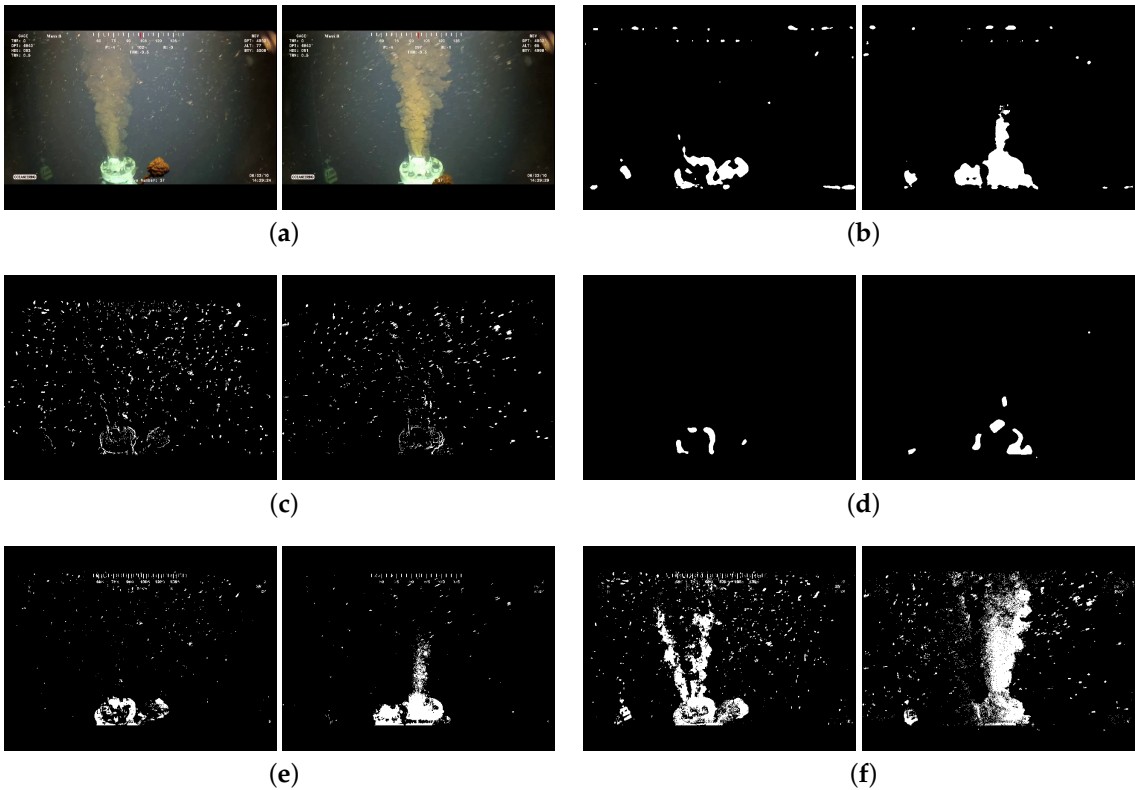

**Figure 4.** Foreground detection: (**a**) original frame; (**b**) LOBSTER [26]; (**c**) MOG2 [17]; (**d**) SuBSENSE [27]; (**e**) VIBE [28]; and (**f**) T2FGMG-UV.

Figure 5 shows the result of color threshold screening. Figure 5a is the original image, which is converted to HSV color space, as shown in Figure 5b, and Figure 5c shows the color mask obtained after threshold screening. The experimental results show that the algorithm can screen out accurate suspected oil spill region, but there is still false detection.

The mask of the suspected oil spill region is obtained by the motion mask bitwise and color mask, as shown in Figure 6a. Directly extracted detection results of the suspected oil spill region mask show that there are still some false detection boxes, as shown in Figure 6b. Figure 6c shows the cell stacks after removing false detection by using STC.

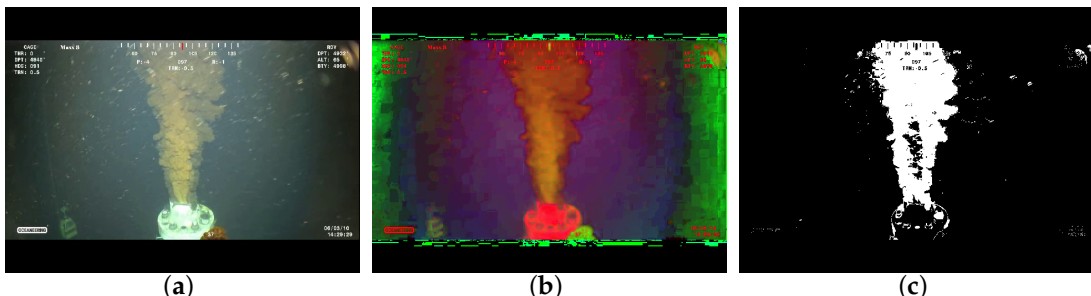

**Figure 5.** HSV threshold: (**a**) original image; (**b**) HSV image; and (**c**) color mask.

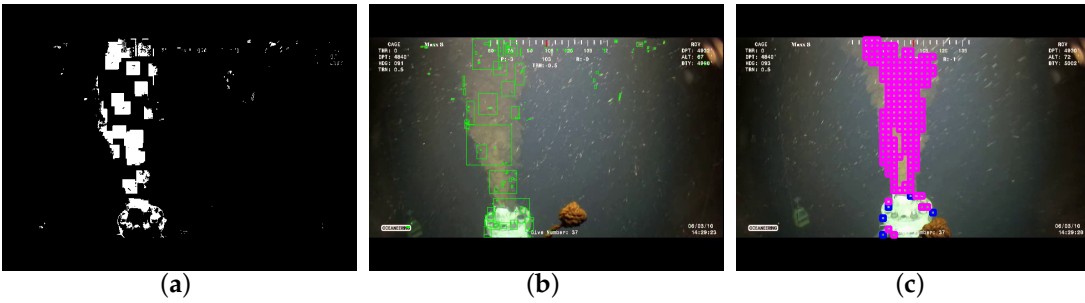

**Figure 6.** detection results: (**a**) fusion-mask; (**b**) pre-detection; and (**c**) cell stacks.

Figure 7a gives the proposed algorithm detection results. Compared to the frame difference in Figure 7b, our results locate the oil spill position and estimate the false detection. Figure 7c shows the yolov3 results. When the iteration number is low, there are many false detections, while, when the iteration number is high, the model is overfitting and it cannot detect oil spill.

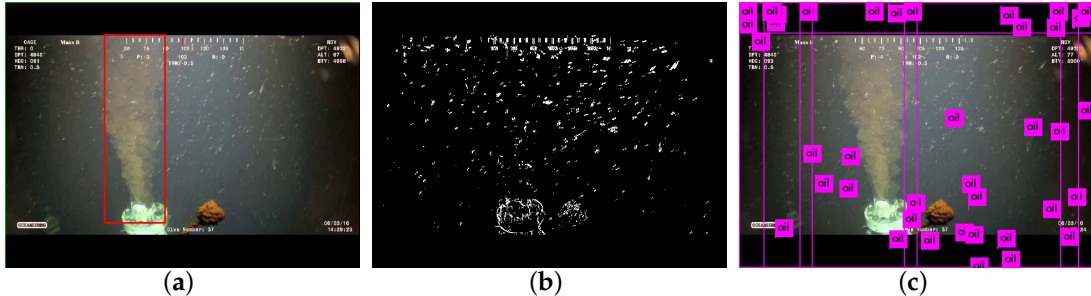

**Figure 7.** Results comparison: (**a**) ours; (**b**) frame difference  [8]; and (**c**) yolov3 [29].

To evaluate algorithm results, we adopted the calculation method of VOC2010 [10]. It includes two import evaluation indicators: precision and recall. Their calculation methods are shown in Equations (12) and (13), respectively.

$$P = \frac{TP}{TP + FP} \tag{12}$$

$$R = \frac{TP}{TP + FN} \tag{13}$$

where TP, FP and TN mean the amount of true positive, false positive and false negative, respectively. They are decided by the intersection over union (IoU) threshold of the detected bounding box and ground truth.

To calculate average precision, the precision at each recall level r is interpolated by taking the maximum precision measured for a method for which the corresponding recall exceeds *r*. The interpolation method is shown in Equation (14). Then, we  computed the average precision (AP) as the area under this curve (AUC) by numerical integration, as shown in Equation (15) .

$$P_{interp(r)} = \max_{\tilde{r}:\tilde{r} \geq r} p(r) \tag{14}$$

$$AP = \sum_{r=0}^{r_{max}} P_{interp(r)} \tag{15}$$

The precision–recall (P-R) curve and average precision (AP) value of the test results are shown in Figure 8 and Table 1. The proposed algorithm achieves extremely high detection accuracy when the intersection over union (IoU) is set at 0.5. However, when the IoU is greater than 0.7, the accuracy of

the proposed algorithm declines, which is due to the limitation of the algorithm caused by the motion detection algorithm and partial oil spill color is close to the color of seawater.

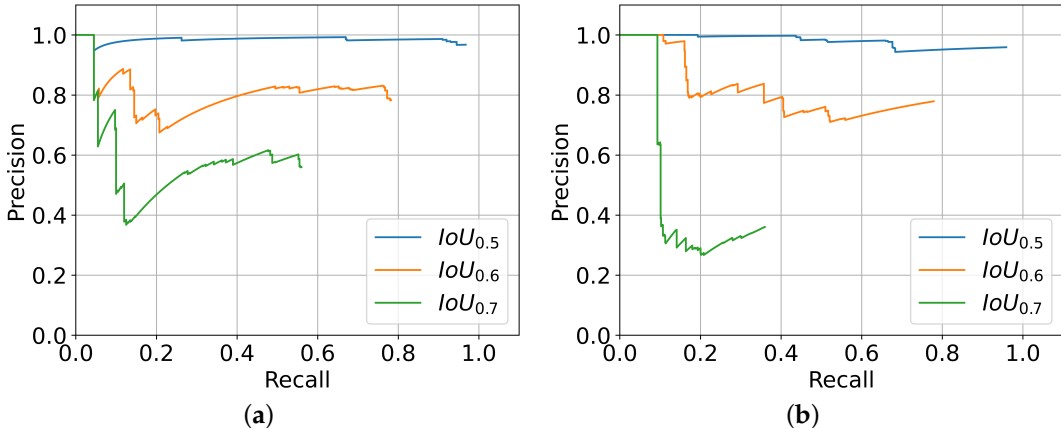

**Figure 8.** precision-recall graph: (**a**) Video 1; (**b**) Video 2.

**Table 1.** average precision.

| Video | $AP_{0.5}$ | $AP_{0.6}$ | $AP_{0.7}$ |
|:-----:|:-----:|:-----:|:-----:|
| 1 | 0.958 | 0.662 | 0.369 |
| 2 | 0.943 | 0.659 | 0.192 |

## 4. Conclusions

This paper presents a method to detect oil spills by underwater images, which can be applied to fixed underwater cameras or remote operated vehicle (ROV). Compared with existing algorithms, it can process more complex real underwater images. It needs less hardware support, thus it can be integrated into an embedded system for independent use with ROVs in the future. It has massive potential for further treatment of the spill area too, such as calculation of spill speed and volume. It has the characteristics of low cost and good timeliness.

The experimental results show that the proposed algorithm can be applied to the alarm operation with less strict requirements for the oil spill position.

However, when the IoU is greater than 0.7, the limitations of the pre-detection part of the algorithm leads to a decline in the proposed algorithm's accuracy. Therefore, further optimization is needed in the detection applications with higher requirements for location and area.

**Author Contributions:** Conceptualization, H.S. and J.S.; methodology, H.S.; software, J.S.; validation, H.S. and J.S.; formal analysis, J.S. and P.R.; writing—original draft preparation, H.S. and J.S.; and writing—review and editing, J.S. and P.R. All authors have read and agreed to the published version of the manuscript.

**Funding:** This research was funded by National Natural Science Foundation of China grant number No. 61602517, the Fundamental Research Funds for the Central Universities( grant No. 18CX02109A), Science and Technology on Electronic Test & Measurement Laboratory Open fund (grant No. 6142001180514) and the Key Program of Marine Economy Development Special Foundation of Department of Natural Resources of Guangdong Province (GDNRC [2020]012)

**Conflicts of Interest:** The authors declare no conflict of interest.

## Abbreviations

The following abbreviations are used in this manuscript:

| | | | |
|---|---|---|---|
| HSV | Hue, Saturation, Value | STC | spatiotemporal context |
| CNN | convolutional neural networks | VOC | visual object classes |
| COCO | common objects in context | SSD | single shot multibox detector |
| YOLO | you only look once | T2FMOG | type-2 fuzzy mixture of Gaussian model |
| MF | membership function | FOU | footprint of uncertainty |
| RGB | red, green, blue | ROV | remote operated vehicle |
| P-R | precision-recall | AP | average precision |
| IoU | intersection over union | AUC | area under curve |

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
