# Peer review of "Underwater Pipeline Oil Spill Detection Based on Structure of Root and Branch Cells"

_jmse, doi:10.3390/jmse8121016_

Round 1
Reviewer 1 Report
The paper addresses an important problem in submarine vision through color restoration, foreground detection and color screening. However, the writing doesn‘t flow where the ideas don‘t progress from one to another. A clear flowchart showing the overall process should be inserted.
For underwater image restoration, there is a lot of previous work that just can be applied to your images. You should focus on the topic of plume-detection in your paper.
The paper is lacking an introduction regarding related work in this field. There are various papers with competing approaches worth mentioning. Refer for example to:
Dhont, D., Jatiault, R., Lattes, P. Combined Use of SAR and Underwater Gliders for Oil Seeps Detection (2019) International Geoscience and Remote Sensing Symposium (IGARSS), art. no. 8898602, pp. 7865-7868. https://www.scopus.com/inward/record.uri?eid=2-s2.0-85077678561&doi=10.1109%2fIGARSS.2019.8898602&partnerID=40&md5=bc2999ab9d06d777153d6780cfbe0ed6 DOI: 10.1109/IGARSS.2019.8898602 Bescond, C., Kruger, S.E., Levesque, D., Brosseau, C. Photoacoustic detection and monitoring of oil spill (2019) AIP Conference Proceedings, 2102, art. no. 020025, . https://www.scopus.com/inward/record.uri?eid=2-s2.0-85066108728&doi=10.1063%2f1.5099729&partnerID=40&md5=3b9d57df5ed2cf7f9b1cf81725a239b5 DOI: 10.1063/1.5099729 Osman, A.B., Ovinis, M., Hashim, F.M., Faye, I. Wavelet-based optical velocimetry for oil spill flow rate estimation (2019) Measurement: Journal of the International Measurement Confederation, 138, pp. 485-496. https://www.scopus.com/inward/record.uri?eid=2-s2.0-85062278787&doi=10.1016%2fj.measurement.2019.01.100&partnerID=40&md5=0b21b11db1a15b4ce83062164e1ba568 DOI: 10.1016/j.measurement.2019.01.100 Qasem, F., Susilo, T.B., Said, S., Alarbash, Z., Hasan, M., Jabakhanji, B., Beyrouthy, T., Alkork, S. Preliminary engineering implementation on multisensory underwater remotely operated vehicle (rov) for oil spills surveillance (2019) BioSMART 2019 - Proceedings: 3rd International Conference on Bio-Engineering for Smart Technologies, art. no. 8734217, . https://www.scopus.com/inward/record.uri?eid=2-s2.0-85068348420&doi=10.1109%2fBIOSMART.2019.8734217&partnerID=40&md5=a97e1b14af3664aa1fdfa429f54a3c73 DOI: 10.1109/BIOSMART.2019.8734217 Zhao, L., Shaffer, F., Robinson, B., King, T., D'Ambrose, C., Pan, Z., Gao, F., Miller, R.S., Conmy, R.N., Boufadel, M.C. Underwater oil jet: Hydrodynamics and droplet size distribution (2016) Chemical Engineering Journal, 299, pp. 292-303. https://www.scopus.com/inward/record.uri?eid=2-s2.0-84967329168&doi=10.1016%2fj.cej.2016.04.061&partnerID=40&md5=50604c30156f72a0c07a889d0d431b18 DOI: 10.1016/j.cej.2016.04.061 Conmy, R.N., Coble, P.G., Farr, J., Wood, A.M., Lee, K., Pegau, W.S., Walsh, I.D., Koch, C.R., Abercrombie, M.I., Miles, M.S., Lewis, M.R., Ryan, S.A., Robinson, B.J., King, T.L., Kelble, C.R., Lacoste, J. Submersible optical sensors exposed to chemically dispersed crude oil: Wave tank simulations for improved oil spill monitoring (2014) Environmental Science and Technology, 48 (3), pp. 1803-1810. https://www.scopus.com/inward/record.uri?eid=2-s2.0-84893561339&doi=10.1021%2fes404206y&partnerID=40&md5=f2bf178b738babd4ef35172e5670d16e DOI: 10.1021/es404206y Oh, S., Lee, M., Seo, S., Roy, M., Seo, D., Kim, J. Underwater multispectral imaging system for environmental monitoring (2014) OCEANS 2014 - TAIPEI, art. no. 6964485, . https://www.scopus.com/inward/record.uri?eid=2-s2.0-84918513777&doi=10.1109%2fOCEANS-TAIPEI.2014.6964485&partnerID=40&md5=88869e8d5adb61da392de487556e4b58 DOI: 10.1109/OCEANS-TAIPEI.2014.6964485 Mooradian, G., Richter, D., Marttila, E.J., Solonenko, M., Saade, E. Spectral Fluorescence/Reflectance optical sensor systems for Arctic oil spill detection and mapping (2014) Society of Petroleum Engineers - Arctic Technology Conference 2014, pp. 409-418. https://www.scopus.com/inward/record.uri?eid=2-s2.0-84904730249&doi=10.4043%2f24590-ms&partnerID=40&md5=56f5bab0a1ba11d1e79e558901d12614 DOI: 10.4043/24590-msReferences should be revised: the mentioned U-Net reference „Akeret, J.; Chang, C.; Lucchi, A.; Refregier, A. Radio frequency interference mitigation using deep convolutional neural networks. Astronomy and Computing 2017, 18, 35–39.“ doesn‘t truly refer to the UNet article:
„Ronneberger O., Fischer P., Brox T. (2015) U-Net: Convolutional Networks for Biomedical Image Segmentation. In: Navab N., Hornegger J., Wells W., Frangi A. (eds) Medical Image Computing and Computer- Assisted Intervention – MICCAI 2015. MICCAI 2015. Lecture Notes in Computer Science, vol 9351. Springer, Cham. https://doi.org/10.1007/978-3-319-24574-4_28“.
The paper has redundancy and unnecessary content (excessively re-writing formulas and content from other papers). Please either provide frames from videos with background (not plain background) and proof that your
method works well, OR state the weaknesses of your method.
Your conclusion should contain an honest summary of what you have reached with your research (referencing related work to be aded in the indtoduction) and also looking on the potential of the approach and derived future work (e.g. could you measure the amount of oil).
Carefully check English language (eg. mistages in figure caption 1 and 7) and correct insertion of blanks.
Reviewer 2 Report
This research is interesting and relevant to important problems involving oil spills. I offer the following suggestions for improvement:
The paper needs to be edited carefully to remove grammar and spacing problems. There are many places where spaces are missing or misplaced. There are also problems with plurals and missing words.
The introduction would benefit from more clarification on where this research fits into the existing literature and what exactly the contribution of this study is. Are you the first researchers to apply these methods to the detection of oil spills?
In the first paragraph of the introduction, indicate the units for the losses of "tens of billions." What currency does this refer to? It would also be good to indicate the number of recent disasters and their annual frequency.
The sentence before reference [6] closely resembles the corresponding sentence in the source. I would suggest paraphrasing that sentence.
In the first sentence under equation (5), I would indicate by whom it is verified by experiments that k=0.9. Is that verified by the present research? If not, who verified it?
In the first sentence in section 2.3, you say, "After a lot of observation, ...." Specify the number of observations and who made these observations.
The images are interesting and compelling. Nice work.
Reviewer 3 Report
This manuscript provides a valuable method for prevention of oil spills. Similar methods should be described, with attention what is new in the presented method and what is adapted or adjusted from the previous works.
Before accepted, some issues should be solved (style, expressions, etc.). Here will be listed only some:
- Typographical errors and style should be corrected through the text. For example, in Abstract, “can’t” should be “cannot” as abbreviations in such manner should not be used in a scientific paper. There are more such errors which can be solved during final reading of the manuscript.
- What is “HSV”?
- Please explain in more details or remove from Abstract “When IoU is 0.5, the average precision can reach more than 0.9.”
- Try to avoid strange units such as “200 million gallons”. Please use SI units.
- At the beginning of Introduction, there is a short note about offshore oil accidents. Regarding that, note: https://doi.org/10.3390/jmse8080555
- What do you mean with “Øistein Johansen’s regard”? This is not proper style for citation in a scientific article.
- Please explain in more details “public data sets such as Pascal VOC, MSCOCO, ImageNet”.
- Try to avoid abbreviations, especially in Conclusion such as “P-R curve and AP”.
- Provide DOI links for references wherever possible.
Round 2
Reviewer 1 Report
Your revision only improved the reviewrs's feedback regarding the langauge an d format and corrected local mistakes. Even there I see room for improvement:
1. Format of the paper should still be improved, such as: line 31, Figure 5 (g), line 332, indentation and spacing ..
2. Authors should follow a certain citation style: e.g. APA style Name et al., 2013 … Lines 43-50.
3. English and wrong spellings should be further improved, such as: line 43 (Bescond propoesd), lines 103-104, line 107 ..
4. Some abbreviations are missing in section Abbreviations line 361, such as: RGHS line 299
More important, the fundamental points of criticism have not been addressed at all in this new version:
- Desription of elated work in oil spill detection and your contribution to the state of the art.
- Neglect the image restoration, which seems to be not in the core of your work (and not as good as related work), and focus on your novel detection method.
- Avoidance of redundancy and unnecessary content.
